# Supervised learning and model analysis with compositional data

**Shimeng Huang**[1]*, **Elisabeth Ailer**[2], **Niki Kilbertus**[2,3], **Niklas Pfister**[1]

**1** Department of Mathematical Sciences, University of Copenhagen, Copenhagen, Denmark, **2** Helmholtz Munich, Munich, Germany, **3** Technical University of Munich, Munich, Germany

\* shimeng@math.ku.dk

**Data Availability Statement:** All data analyzed in this paper has been previously published and is publicly available online. Details of data sources and preprocessing steps can be found in the appendix. All experiments are reproducible using

## Abstract

Supervised learning, such as regression and classification, is an essential tool for analyzing modern high-throughput sequencing data, for example in microbiome research. However, due to the compositionality and sparsity, existing techniques are often inadequate. Either they rely on extensions of the linear log-contrast model (which adjust for compositionality but cannot account for complex signals or sparsity) or they are based on black-box machine learning methods (which may capture useful signals, but lack interpretability due to the compositionality). We propose `KernelBiome`, a kernel-based nonparametric regression and classification framework for compositional data. It is tailored to sparse compositional data and is able to incorporate prior knowledge, such as phylogenetic structure. `KernelBiome` captures complex signals, including in the zero-structure, while automatically adapting model complexity. We demonstrate on par or improved predictive performance compared with state-of-the-art machine learning methods on 33 publicly available microbiome datasets. Additionally, our framework provides two key advantages: (i) We propose two novel quantities to interpret contributions of individual components and prove that they consistently estimate average perturbation effects of the conditional mean, extending the interpretability of linear log-contrast coefficients to nonparametric models. (ii) We show that the connection between kernels and distances aids interpretability and provides a data-driven embedding that can augment further analysis. `KernelBiome` is available as an open-source Python package on PyPI and at https://github.com/shimenghuang/KernelBiome.

## Author summary

In recent years, advances in gene sequencing technology have allowed scientists to examine entire microbial communities within genetic samples. These communities interact with their surroundings in complex ways, potentially benefiting or harming the host they inhabit. However, analyzing the microbiome—the measured microbial community—is challenging due to the compositionality and sparsity of the data.

In this study, we developed a statistical framework called `KernelBiome` to model the relationship between the microbiome and a target of interest, such as the host's disease status. We utilized a type of machine learning model called kernel methods and adapted

the code in https://github.com/shimenghuang/KernelBiome.

**Funding:** SH and NP are supported by a research grant (0069071) from Novo Nordisk Fonden. EA is supported by the Helmholtz Association under the joint research school "Munich School for Data Science - MUDS". The funders had no role in study design, data collection and analysis, decision to publish, or preparation of the manuscript.

**Competing interests:** The authors have declared that no competing interests exist.

them to handle the compositional and sparse nature of the data, while also incorporating prior expert knowledge.

Additionally, we introduced two new measures to help interpret the contributions of individual compositional components. Our approach also demonstrated that kernel methods increase interpretability in analyzing microbiome data. To make `KernelBiome` as accessible as possible, we have created an easy-to-use software package for researchers and practitioners to apply in their work.

## 1 Introduction

Compositional data, that is, measurements of parts of a whole, are common in many scientific disciplines. For example, mineral compositions in geology [1], element concentrations in chemistry [2], species compositions in ecology [3] and more recently high-throughput sequencing reads in microbiome science [4].

Mathematically, any $p$-dimensional composition—by appropriate normalization—can be represented as a point on the simplex

$$\mathbb{S}^{p-1} := \{x \in [0,1]^p \mid \sum_{j=1}^{p} x^j = 1\}.$$

This complicates the statistical analysis, because the sum-to-one constraint of the simplex induces non-trivial dependencies between the components that may lead to false conclusions, if not appropriately taken into account.

The statistics community has developed a substantial collection of parametric analysis techniques to account for the simplex structure. The most basic is the family of Dirichlet distributions. However, as pointed out already by [5], Dirichlet distributions cannot capture non-trivial dependence structures between the composition components and are thus too restrictive. [5] therefore introduced the *log-ratio* approach. It generates a family of distributions by projecting multivariate normal distributions into $\mathbb{S}^{p-1}$ via an appropriate log-ratio transformation (e.g., the additive log-ratio, centered log-ratio [5], or isometric log-ratio [6]). The resulting family of distributions results in parametric models on the simplex that are rich enough to capture non-trivial dependencies between the components (i.e., beyond those induced by the sum-to-one constraint). The log-ratio approach has been extended and adapted to a range of statistical problems [e.g., 7–11].

For supervised learning tasks the log-ratio approach leads to the *log-contrast model* [12]. An attractive property of the log-contrast model is that its coefficients quantify the effect of a multiplicative perturbation (i.e., fractionally increasing one component while adjusting the others) on the response. While several extensions of the log-contrast model exist [e.g., 13–17], its parametric approach to supervised learning has two major shortcomings that become particularly severe when applied to high-dimensional and zero-inflated high-throughput sequencing data [18, 19]. Firstly, since the logarithm is not defined at zero, the log-contrast model cannot be directly applied. A common fix is to add so-called pseudo-counts, a small non-zero constant, to all (zero) entries [20, 21]. More sophisticated replacements exist as well [e.g., 22–24], however, they often rely on knowing the nature of the zeros (e.g., whether they are structural or random), which is typically not available in practice and difficult to estimate. In any case, the downstream analysis will strongly depend on the selected zero imputation scheme [25]. Secondly, the relationships between individual components (e.g., species) and the response are generally complex. For example, in human microbiome settings, a health outcome may

depend on interactions or on the presence or absence of species. Both cannot be captured by the linear structure of the log-contrast model.

We propose to solve the supervised learning task using a nonparametric kernel approach, which is able to handle complex signals and avoid arbitrary zero-imputation. To be of use in biological applications, there are two components to a supervised analysis: (i) estimating a predictive model that accurately captures signals in the data and (ii) extracting meaningful and interpretable information from the estimated model. For (i), it has been shown that modern machine learning methods are capable of creating highly predictive models by using microbiome data as covariates and phenotypes as responses [e.g., 26–29]. In particular, several approaches have been proposed where kernels are used to incorporate prior information [30, 31], as a way to utilize the compositional structure [32–34] and to construct association tests [35–37]. Our proposed framework extends these works by providing new post-analysis techniques (e.g., the compositional feature influence) that respect the compositional structure. Recently, [25, 38] used the radial transformation to argue that kernels on the sphere provide a natural way of analyzing compositions with zeros and similar to our work suggest using the kernel embeddings in a subsequent analysis. Part (ii) is related to the fields of explainable artificial intelligence [39] and interpretable machine learning [40], which focus on extracting information from predictive models. These types of approaches have also received growing attention in the context of microbiome data [41–43]. However, to the best of our knowledge, none of these procedures have been adjusted to account for the compositional structure. As we show in Sec. 2.1, not accounting for the compositionality may invalidate the results.

KernelBiome, see Fig 1, addresses both (i) by providing a regression and classification procedure based on kernels targeted to the simplex and (ii) by providing a principled way of analyzing the estimated models. Our contributions are fourfold: (1) We develop a theoretical framework for using kernels on compositional data. While using kernels to analyze various aspects of compositional data is not a new idea, a comprehensive analysis and its connection to existing approaches has been missing. In this work, we provide a range of kernels that each capture different aspects of the simplex structure, many of which have not been previously applied to compositional data. For all kernels, we derive novel, positive-definite weighted versions that allow incorporating prior information between the components. Additionally, we

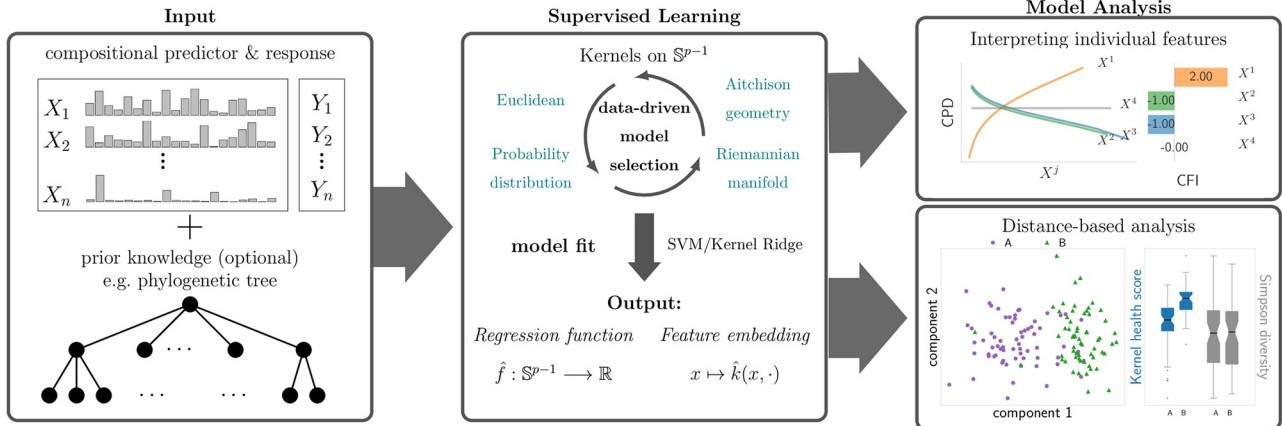

**Fig 1. Overview of KernelBiome.** We start from a paired dataset with a compositional predictor $X$ and a response $Y$ and optional prior knowledge on the relation between components in the compositions (e.g., via a phylogenetic tree). We then select a model among a large class of kernels which best fits the data. This results in an estimated model $\hat{f}$ and embedding $\hat{k}$. Finally, these can be analyzed while accounting for the compositional structure.

show that the distance associated with each kernel can be used to define a kernel-based scalar summary statistic. (2) We propose a theoretically justified analysis of kernel-based models that accounts for compositionality. Firstly, we introduce two novel quantities for measuring the effects of individual features that explicitly take the compositionality into account and prove that these can be consistently estimated. Secondly, we build on known connections between kernels and distance measures to advocate for using the kernel embedding from the estimated model to create visualizations and perform follow-up distance-based analyses that respect the compositionality. (3) We draw connections between `KernelBiome` and log-contrast-based analysis techniques. More specifically, we connect the Aitchison kernel to the log-contrast model, prove that the proposed compositional feature influence in this case reduces to the log-contrast coefficients, and show that our proposed weighted Aitchison kernel is related to the recently proposed tree-aggregation method of log-contrast coefficients [44]. Importantly, these connections ensure that `KernelBiome` reduces to standard log-contrast analysis techniques whenever simple log-contrast models are capable of capturing most of the signal. This is also illustrated by our experimental results. (4) We propose a data-adaptive selection framework that allows to compare different kernels in a principled fashion.

The paper is structured as follows. In Sec. 2, we introduce the supervised learning task, define two quantities for analyzing individual components (Sec. 2.1), give a short introduction to kernel methods and how to apply our methodology (Sec. 2.2), and present the full `KernelBiome` framework (Sec. 2.3). Finally, we illustrate the advantages of `KernelBiome` in the experiments in Sec. 3.

## 2 Methods

We consider the setting in which we observe $n$ independent and identically distributed (i.i.d.) observations $(X_1, Y_1), \ldots, (X_n, Y_n)$ of a random variable $(X, Y)$ with $X \in \mathbb{S}^{p-1}$ a compositional predictor and $Y \in \mathbb{R}$ a real-valued response variable (by which we include categorical responses). Supervised learning attempts to learn a relationship between the response $Y$ and the dependent predictors $X$. In this work, we focus on conditional mean relationships. More specifically, we are interested in estimating the conditional mean of $Y$, that is, the function

$$f^* : x \mapsto \mathbb{E}[Y \mid X = x]. \tag{1}$$

We assume that $f^* \in \mathcal{F} \subseteq \{f \mid f : \mathbb{S}^{p-1} \to \mathbb{R}\}$, where $\mathcal{F}$ is a function class determined by the regression (or classification) procedure.

While estimating and analyzing the conditional mean is well established for predictors in Euclidean space, there are two factors that complicate the analysis when the predictors are compositional. (i) While it is possible to directly apply most standard regression procedures designed for $X \in \mathbb{R}^p$ also for $X \in \mathbb{S}^{p-1}$, it turns out that many approaches are ill-suited to approximate functions on the simplex. (ii) Even if one accurately estimates the conditional mean function $f^*$, the simplex constraint complicates any direct assessment of the influence and importance of individual components of the compositional predictor. In this work, we address both issues and propose a nonparametric framework for regression and classification analysis for compositional data.

### 2.1 Interpreting individual features

Our goal when estimating the conditional mean $f^*$ given in (1) is to gain insight into the relationship between the response $Y$ and predictors $X$. For example, when fitting a log-contrast model (see Example 2.1), the estimated coefficients provide a useful tool to generate hypotheses about which features affect the response and thereby inform follow-up experiments. For

more complex models, such as the nonparametric methods proposed in this work, direct interpretation of a fitted model $\hat{f}$ is difficult. Two widely applicable measures due to [45] are the following: (i) Relative influence, which assigns each coordinate $j$ a scalar influence value given by the expected partial derivative $\mathbb{E}\left[\frac{d}{dx^j}\hat{f}(X)\right]$ and (ii) partial dependence plots, which are constructed by plotting, for each coordinate $j$, the function $z \mapsto \mathbb{E}[\hat{f}(X^1, \ldots, X^{j-1}, z, X^{j+1}, \ldots, X^p)]$. However, directly applying these measures on the simplex is not possible as we illustrate in Fig B in S4 Appendix. The intuition is that both measures evaluate the function $\hat{f}$ outside the simplex. An adaptation of the relative influence (or elasticity in the econometrics literature) to compositions based on the Aitchison geometry has recently been proposed by [46]. We adapt the relative influence without relying on the log-ratio transform and hence allow for more general function classes.

Our approach is based on two coordinate-wise perturbations on the simplex. For any $j \in \{1, \ldots, p\}$ and $x \in \mathbb{S}^{p-1}$, define (i) for $c \in [0, \infty)$ the function $\psi_j(x, c) \in \mathbb{S}^{p-1}$ to be the composition resulting from multiplying the $j$-th component by $c$ and then scaling the entire vector back into the simplex, and (ii) for $c \in [0, 1]$ the function $\phi_j(x, c) \in \mathbb{S}^{p-1}$ to be the composition that consists of fixing the $j$-th coordinate to $c$ and then rescaling all remaining coordinates such that the resulting vector lies in the simplex. Each perturbation can be seen as a different way of intervening on a single coordinate while preserving the simplex structure. More details are given in S1 Appendix. Based on the first perturbation, we define the *compositional feature influence* (CFI) of component $j \in \{1, \ldots, p\}$ for any differentiable function $f : \mathbb{S}^{p-1} \to \mathbb{R}$ by

$$\text{(CFI)} \qquad I_f^j := \mathbb{E}\left[\frac{d}{dc}f(\psi_j(X, c))\big|_{c=1}\right]. \qquad (2)$$

Similarly, we adapt partial dependence plots using the second perturbation. Define the *compositional feature dependence* (CPD) of component $j \in \{1, \ldots, p\}$ for any function $f : \mathbb{S}^{p-1} \to \mathbb{R}$ by

$$\text{(CPD)} \qquad S_f^j : z \mapsto \mathbb{E}[f(\phi_j(X, z))] - \mathbb{E}[f(X)]. \qquad (3)$$

In practice, we can compute Monte Carlo estimates of both quantities by replacing expectations with empirical means. We denote the corresponding estimators by $\hat{I}_f^j$ and $\hat{S}_f^j$, respectively (see S1 Appendix for details).

The following proposition connects the CFI and CPD to the coefficients in a log-contrast function.

**Proposition 2.1** (CFI and CPD in the log-contrast model). Let $f : x \mapsto \beta^T \log(x)$ with $\sum_{j=1}^p \beta_j = 0$, then the CFI and CPD are given by

$$I_f^j = \beta_j \quad \text{and} \quad S_f^j : z \mapsto \beta_j \log\left(\frac{z^j}{1-z^j}\right) + c,$$

respectively, where $c \in \mathbb{R}$ is a constant depending on the distribution of $X$ but not on $z$ and satisfies $c = 0$ if $\beta_j = 0$.

A proof is given in S6 Appendix. The proposition shows that the CFI and CPD are generalizations of the $\beta$-coefficients in the log-contrast model. The following example provides further intuition.

**Example 2.1** (CFI and CPD in a log-contrast model). Consider a log-contrast model $Y = f(X) + \epsilon$ with $f : x \mapsto 2 \log(x^1) - \log(x^2) - \log(x^3)$.

The CFI and CPD for the true function $f$—estimated based on $n = 100$ i.i.d. samples $(X_1, Y_1), \ldots, (X_n, Y_n)$ with $X_i$ compositional log-normal—are shown in Fig 2.

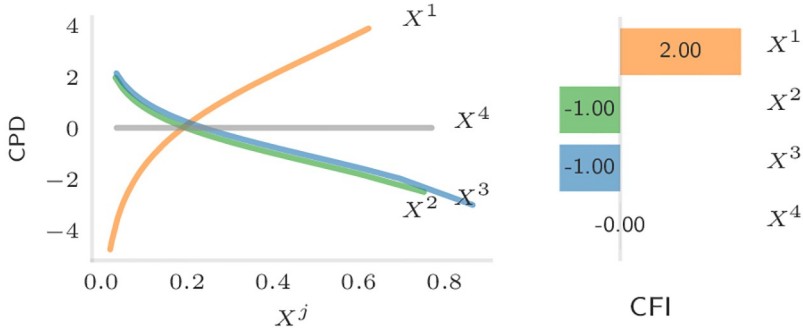

**Fig 2. Visualization of the CPD (left) and CFI (right) based on $n = 100$ samples and the true function $f$.** Since $\beta_4 = 0$ in this example the 4-th component has no effect on the value of $f$ resulting in a CFI of zero and a flat CPD. Since we are not estimating $f$, the CFI values exactly correspond to the $\beta$-coefficients in this example.

The following theorem highlights the usefulness of the CFI and CPD by establishing when they can be consistently estimated from data.

**Theorem 2.1** (Consistency). Assume $\hat{f}_n$ is an estimator of the conditional mean $f^*$ given in (1) based on $(X_1, Y_1), \ldots, (X_n, Y_n)$ i.i.d.

(i) If $\frac{1}{n}\sum_{i=1}^{n} \|\nabla\hat{f}_n(X_i) - \nabla f^*(X_i)\|_2 \xrightarrow{P} 0$ as $n \to \infty$ and $\mathbb{E}[(\nabla f^*(X_i))^2] < \infty$, then it holds for all $j \in \{1, \ldots, p\}$ that

$$\hat{I}^j_{\hat{f}_n} \xrightarrow{P} I^j_{f^*} \quad \text{as } n \to \infty.$$

(ii) If $\sup_{x \in \text{supp}(X)} |\hat{f}_n(x) - f^*(x)| \xrightarrow{P} 0$ as $n \to \infty$ and $\text{supp}(X) = \{w/(\sum_j w^j) | w \in \text{supp}(X^1) \times \cdots \times \text{supp}(X^p)\}$, then it holds for all $j \in \{1, \ldots, p\}$ and all $z \in [0, 1]$ with $z/(1 - z) \in \text{supp}(X^j/\sum_{\ell \neq j} X^\ell)$ that

$$\hat{S}^j_{\hat{f}_n}(z) \xrightarrow{P} S^j_{f^*}(z) \quad \text{as } n \to \infty.$$

A proof is given in S6 Appendix and the result is demonstrated on simulated data in Fig A in S4 Appendix. The theorem shows that the CFI is consistently estimated as long as the derivative of $f^*$ is consistently estimated, which can be ensured for example for the kernel methods discussed in Sec. 2.2. In contrast, the CPD only requires the function $f^*$ itself to be consistently estimated. The additional assumption on the support ensures that the perturbation $\phi_j$ used in the CPD remains within the support. If this assumption is not satisfied one needs to ensure that the estimated function extrapolates beyond the sample support. Interpreting the CPD therefore requires caution.

## 2.2 Kernel methods for compositional data analysis

Before presenting our proposed weighted and unweighted kernels, we briefly review the necessary background on kernels and their connection to distances. Kernel methods are a powerful class of nonparametric statistical methods that are particularly useful for data from non-standard (i.e., non-Euclidean) domains $\mathcal{X}$. The starting point is a symmetric, positive definite function $k : \mathcal{X} \times \mathcal{X} \to \mathbb{R}$, called kernel. Kernels encode similarities between points in $\mathcal{X}$, i.e.,

large values of $k$ correspond to points that are similar and small values to points that are less similar. Instead of directly analyzing the data on $\mathcal{X}$, kernel methods map it into a well-behaved feature space $\mathcal{H}_k \subseteq \{f \mid f : \mathcal{X} \to \mathbb{R}\}$ called reproducing kernel Hilbert space (RKHS), whose inner product preserves the kernel induced similarity.

Here, we consider kernels on the simplex, that is, $\mathcal{X} = \mathbb{S}^{p-1}$. The conditional mean function $f^*$ given in (1) can then be estimated by optimizing a loss over $\mathcal{H}_k$, for an appropriate kernel $k$ for which $\mathcal{H}_k$ is sufficiently rich, i.e., $f^* \in \mathcal{H}_k$. The representer theorem [e.g., 47] states that such an optimization over $\mathcal{H}_k$ can be performed efficiently. Formally, it states that the minimizer of an arbitrary convex loss function $L : \mathbb{R}^n \times \mathbb{R}^n \to [0, \infty)$ of the form

$$\hat{f} = \underset{f \in \mathcal{H}_k}{\arg\min} L((Y_1, f(X_1)), \ldots, (Y_n, f(X_n))) + \lambda \|f\|^2_{\mathcal{H}_k},$$

with $\lambda > 0$ a penalty parameter, has the form $\hat{f}(\cdot) = \sum_{i=1}^n \hat{\alpha}_i k(X_i, \cdot)$ for some $\hat{\alpha} \in \mathbb{R}^n$. This means that instead of optimizing over a potentially infinite-dimensional space $\mathcal{H}_k$, it is sufficient to optimize over the $n$-dimensional parameter $\hat{\alpha}$. Depending on the loss function, this allows to construct efficient regression and classification procedures, such as kernel ridge regression and support vector machines [e.g., 47].

The performance of the resulting prediction model depends on the choice of kernel as this determines the function space $\mathcal{H}_k$. A useful way of thinking about kernels is via their connection to distances. In short, any kernel $k$ induces a unique semi-metric $d_k$ and vice versa. More details are given in S5 Appendix. This connection has two important implications. Firstly, it provides a natural way for constructing kernels based on established distances on the simplex. The intuition being that a distance, which is large for observations with vastly different responses and small otherwise, leads to an informative feature space $\mathcal{H}_k$. Secondly, it motivates using the kernel-induced distance, see Sec. 2.3.2.

**2.2.1 Kernels on the simplex.**   We consider four types of kernels on the simplex, each related to different types of distances. A full list with all kernels and induced distances is provided in S7 Appendix. While most kernels have previously appeared in the literature, we have adapted many of the kernels to fit into the framework provided here, e.g., added zero-imputation for Aitchison kernels and updated the parametrization for the probability distribution kernels.

**Euclidean**. These are kernels that are constructed by restricting kernels on $\mathbb{R}^p$ to the simplex. Any such restriction immediately guarantees that the restricted kernel is again a kernel. However, the induced distances are not targeted to the simplex and therefore can be unnatural choices. In KernelBiome, we have included the linear kernel and the radial basis function (RBF) kernel. The RBF kernel is $L^p$-universal [e.g., 48] which means that it can approximate any integrable function (in the large sample limit). However, this does not necessarily imply good performance for finite sample sizes.

**Aitchison geometry**. One way of incorporating the simplex structure is to use the Aitchison geometry. Essentially, this corresponds to mapping points from the interior of the simplex via the centered log-ratio transform into $\mathbb{R}^p$ and then using the Euclidean geometry. This results in the Aitchison kernel for which the induced RKHS is equal to the log-contrast functions. In particular, applying kernel ridge regression with an Aitchison kernel corresponds to fitting a log-contrast model with a penalty on the coefficients. As the centered log-ratio transform is only defined for interior points in the simplex, we add a hyperparameter to the kernels that shift them away from zero. From this perspective, the commonly added pseudo-count constant added to all components becomes a tuneable hyperparameter of our method, rather than a fixed ad-hoc choice during data pre-processing. Thereby, our modified Aitchison kernel

respects the fact that current approaches to zero-replacement or imputation are often not biologically justified, yet may impact predictive performance. Our proposed zero-imputed Aitchison kernel comes with two advantages over standard log-contrast modelling: (1) A principled adjustment for zeros and (2) an efficient form of high-dimensional regularization that performs well across a large range of our experiments. In KernelBiome, we include the Aitchison kernel and the Aitchison-RBF kernel which combines the Aitchison and RBF kernels.

**Probability distributions**. Another approach to incorporate the simplex structure into the kernel is to view points in the simplex as discrete probability distributions. This allows us to make use of the extensive literature on distances between probability distributions to construct kernels. In KernelBiome, we have adapted two classes of such kernels: (1) A parametric class based on generalized Jensen-Shannon distances due to [49], which we call generalized-JS, and (2) a parametric class based on the work by [50], which we call Hilbertian. Together they contain many well-established distances such as the total variation, Hellinger, Chi-squared, and Jensen-Shannon distance. All resulting kernels allow for zeros in the components of compositions.

**Riemannian manifold**. Finally, the simplex structure can be incorporated by using a multinomial distribution which has a parameter in the simplex. [51] show that the geometry of multinomial statistical models can be exploited by using kernels based on the heat equation on a Riemannian manifold. The resulting kernel is known as the heat-diffusion kernel and has been observed to work well with sparse data.

**2.2.2 Including prior information into kernels.** All kernels introduced in the previous section (and described in detail in S7 Appendix) are invariant under permutations of the compositional components. They therefore do not take into account any relation between the components. In many applications, one may however have prior knowledge about the relation between the components. For example, if the compositional predictor consists of relative abundances of microbial species, information about the genetic relation between different species encoded in a phylogenetic tree may be available. Therefore, we provide the following way to incorporate such relations. Assume the prior information has been expressed as a positive semi-definite weight matrix $W \in \mathbb{R}^{p \times p}$ with non-negative entries (e.g., using the UniFrac-Distance [52] as shown in Sec. C of S2 Appendix), where the $ij$-th entry corresponds to the strength of the relation between components $i$ and $j$. We can then incorporate $W$ directly into our kernels. To see how this works, consider the special case where the kernel $k$ can be written as $k(x, y) = \sum_{i=1}^{p} k_0(x^i, y^i)$ for a positive definite kernel $k_0 : [0, 1] \times [0, 1] \to \mathbb{R}$. Then, the weighted kernel

$$k_W(x, y) := \sum_{i,j=1}^{p} W_{i,j} \cdot k_0(x^i, y^j) \tag{4}$$

is positive definite and incorporates the prior information in a natural way. If two components $i$ and $j$ are known to be related (corresponding to large values of $W_{i,j}$), the kernel $k_W$ takes the similarity across these components into account. In Sec. B of S2 Appendix, we show that the probability distribution kernels and the linear kernel can be expressed in this way and propose similar weighted versions for the remaining kernels.

An advantage of our framework is that it defaults to the log-contrast model when more complex models fail to improve the prediction (due to the zero-shift in our proposed Aitchison kernel and the kernel-based regularization, this correspondence is however not exact). We now show that for the weighted Aitchison kernel, the RKHS consists of log-contrast functions with equal coefficients across the weighted blocks, this is similar to how [44] incorporate prior information into log-contrast models.

**Proposition 2.2** (weighted Aitchison kernel RKHS). Let $P_1, \ldots, P_m \subseteq \{1, \ldots, p\}$ be a disjoint partition and $W \in \mathbb{R}^{p \times p}$ the weight matrix defined for all $i, j \in \{1, \ldots, p\}$ by $W_{i,j} := \sum_{\ell=1}^{m} \frac{1}{|P_\ell|} \mathbb{1}_{\{i,j \in P_\ell\}}$. Let $k_W$ be the weighted Aitchison kernel given in S7 Appendix (but without zero imputation and on the open simplex). Then, it holds that

$$f \in \mathcal{H}_{k_W} \quad \Leftrightarrow \quad f = \beta^\top \log(\cdot) \tag{7}$$

for some $\beta \in \mathbb{R}^p$ satisfying (1) $\sum_{j=1}^{p} \beta_j = 0$ and (2) for all $\ell \in \{1, \ldots, m\}$ and $i, j \in P_\ell$ it holds $\beta_i = \beta_j$.

A proof is given in S6 Appendix. Combined with Proposition 2.1, this implies that the CFI values are equal across the equally weighted blocks $P_1, \ldots, P_m$, which is demonstrated empirically in Sec. 3.2 and Sec. 3.4.

## 2.3 KernelBiome framework

For a given i.i.d. dataset $(X_1, Y_1), \ldots, (X_n, Y_n)$, `KernelBiome` first runs a data-driven model selection, resulting in an estimated regression function $\hat{f}$ and a specific kernel $\hat{k}$ (see Fig 1). Then, the feature influence properties (CFI, CPD) and embedding induced by $\hat{k}$ are analyzed in a way that respects compositionality.

**2.3.1 Model selection.**   We propose the following two step data-driven selection procedure.

1. Select the best kernel $\hat{k}$ with the following hierarchical CV:

   - Fix a kernel $\tilde{k}$, i.e., a type of kernel and its kernel parameters.

   - Split the sample into $N_{\text{out}}$ random (or stratified) folds.

   - For each fold, use all other folds to perform a $N_{\text{in}}$-fold CV to select the best hyperparameter $\tilde{\lambda}$ and compute a CV score based on $\tilde{k}$ and $\tilde{\lambda}$ on the left-out fold.

   - Select the kernel $\hat{k}$ with the best average CV score.

2. Select the best hyperparameter $\hat{\lambda}$ for $\hat{k}$ using a $N_{\text{in}}$-fold CV on the full data. The final estimator $\hat{f}$ is then given by the kernel predictor based on $\hat{k}$ and $\hat{\lambda}$.

This CV scheme ensures that all parameters are selected in a data adaptive way. Up to a point, including more parameter settings into the CV makes the method more robust at the cost of additional run time. We provide sensible default choices for all parameters (see e.g., Table A in Sec. A of S2 Appendix for the default kernels), allowing practitioners to directly apply the method. In the `KernelBiome` implementation, the parameter grids for the kernel parameters and hyperparameters, as well as parameters of the CV including the type of CV, number of CV folds, and scoring can also be be adjusted manually, for example to reduce the run time.

**2.3.2 Model analysis.**   Firstly, as discussed in Sec. 2.1, we propose to analyze the fitted model $\hat{f}$ with the CPD and CFI. Other methods developed for functions on $\mathbb{R}^p$ do not account for compositionality and can be misleading. Secondly, the kernel embedding $\hat{k}$ can be used for the following two types of analyses.

**Distance-based analysis**. A key advantage of using kernels is that the fitted kernel $\hat{k}$ is itself helpful in the analysis. As discussed in Sec. 2.2, $\hat{k}$ induces a distance on the simplex that is well-suited to separate observations with different response values. We therefore

suggest to utilize this distance to investigate the data further. Essentially, any statistical method based on distances can be applied. We specifically suggest using kernel PCA to project the samples into a two-dimensional space. As we illustrate in Sec. 3.3, such a projection can be used to detect specific groups or outliers in the samples and can also help understand how the predictors are used by the prediction model $\hat{f}$. As we are working with compositional data we need to be careful when looking at how individual components contribute to each principle component. Fortunately, the perturbation $\psi$ defined in Sec. 2.1 can again be used to construct informative component-wise measures. All details on kernel PCA and how to compute component-wise contributions for each principle component are provided in Sec. B of S5 Appendix.

**Data-driven scalar summary statistics**. Practitioners often work with scalar summaries of the data as these are easy to communicate. A commonly used summary statistic in ecology is $\alpha$-diversity which measures the variation within a community. The connection between kernels and distances provides a useful tool to construct informative scalar summary statistics by considering distances to a reference point $u$ in the simplex. Formally, for a fixed reference point $u \in \mathbb{S}^{p-1}$ define for all $x \in \mathbb{S}^{p-1}$ a corresponding closeness measure by $D^k(x) := -d_k^2(x, u)$, where $d_k$ is the distance induced by the kernel $k$. This provides an easily interpretable scalar quantity. For example, if $Y$ is a binary indicator taking values *healthy* and *sick*, we could select $u$ to be the geometric median (the observation that has the smallest total distance to all the other observations based on the pairwise kernel distance) of all $X_i$ with $Y^i =$ *healthy*. Then, $D^k$ corresponds to a very simple health score (see Sec. 3.3 for a concrete example). A further example is given by selecting $u = (1/p, \ldots, 1/p)$ and considering points on the simplex as communities. Then, $u$ can be interpreted as the most diverse point in the simplex and $D^k$ corresponds to a data-adaptive $\alpha$-diversity measure. While such a definition of diversity does not necessarily satisfy all desirable properties for diversities [see e.g., 53], it is (1) symmetric with respect to switching of coordinates, (2) has an intuitive interpretation and (3) is well-behaved when combined with weighted kernels. Connections to established diversities also exist, for example, the linear kernel corresponds to a shifted version of the Gini-Simpson diversity (i.e., $\text{Gini} - \text{Simpson}(x) := 1 - \sum_{j=1}^{p} (x^j)^2 = D^k(x) + \frac{p-2}{p}$).

**2.3.3 Run time complexity.** The run time complexity of `KernelBiome` depends on the number of kernels $K$, the size of the hyperparameter grid $H$, and the number of inner CV folds $N_{\text{in}}$ and outer CV folds $N_{\text{out}}$. Since the run time complexity of kernel ridge regression and support vector machines is $O(n^3)$ (based on a straightforward implementation, actual implementation in available software libraries can achieve a more optimized run time), the total run time complexity of `KernelBiome` is $O(KHN_{\text{in}}N_{\text{out}}n^3)$. For example, the default parameter settings use 55 kernels, with 5-fold inner CV and 10-fold outer CV with a hyperparameter grid of size 10, resulting in 27,500 model fits, each of complexity $O(n^3)$. To reduce the run time we recommend reducing the number of kernels $K$, this can be particularly useful for prototyping. However, if possible, we recommend using the full list of kernels for a final analysis to avoid a decrease in predictive performance.

**2.3.4 Implementation.** `KernelBiome` is implemented as a Python [54] package that takes advantage of the high-performance JAX [55] and `scikit-learn` [56] libraries. All kernels introduced are implemented with JAX's just-in-time compilation and automatically leverage accelerators such as GPU and TPU whenever available. `KernelBiome` provides fast computation of all kernels and distance metrics as well as easy-to-use procedures for model selection and comparison and procedures to estimate CPD and CFI, compute kernel PCA, and estimate scalar summary statistics. An illustration script for the package's usage can be found in the package repository.

## 3 Results

We evaluated `KernelBiome` on a total of 33 microbiome datasets. All datasets have been previously published and final datasets used in our experiments can be fully reproduced following the description in the GitHub repo https://github.com/shimenghuang/KernelBiome. A summary of the datasets including on the pre-processing steps, prediction task and references is provided in Table A in S3 Appendix. First, in Sec. 3.1, we show that `KernelBiome` performs on par or better than existing supervised learning procedures for compositional data, while reducing to a powerfully regularized version of the log-contrast model if the prediction task is simple. In Sec. 3.2, we show on a semi-artificial dataset that including prior information can either improve or harm the predictive performance depending on whether or not it is relevant for the prediction. In Sec. 3.3, we illustrate the advantages of a full analysis with `KernelBiome`. Finally, in Sec. 3.4, we demonstrate how `KernelBiome` can incorporate prior knowledge, while preserving a theoretically justified interpretation.

### 3.1 State-of-the-art prediction performance

We compare the predictive performance of `KernelBiome` on all datasets with the following competitors: (i) `Baseline`, a naive predictor that uses the training majority class for classification and the training mean for regression as its prediction, (ii) `SVM-RBF`, a support vector machine with the RBF kernel, (iii) `Lin/Log-L1`, a linear/logistic regression with $\ell^1$-penalty (iv) `LogCont-L1`, a log-contrast regression with $\ell^1$ penalty with a half of the minimum non-zero relative abundance added as pseudo-count to remove zeros, and (v) `RF`, a random forest with 500 trees. For `SVM-RBF`, `Lin/Log-L1` and `RF` we use the `scikit-learn` implementations [56] and choose the hyperparameters (bandwidth, max depth and all penalty parameters) based on a 5-fold CV. For `LogCont-L1`, we use the `c-lasso` package [16] and the default CV scheme to chose the penalty parameter. We apply two versions of `KernelBiome`: (1) The standard version that adaptively chooses the kernel using $N_{in} = 5$, $N_{out} = 10$ (denoted `KernelBiome`), and (2) a version with fixed Aitchison kernel with $c$ equal to half of the minimum non-zero relative abundance (denoted `KB-Aitchison`). Both methods use a default hyperparameter grid of size 40, and we use kernel ridge regression as the estimator. We compared with using support vector regression instead of kernel ridge regression and the results are similar.

For the comparison we perform 20 random (stratified) 10-fold train/test splits and record the predictive performance (balanced accuracy for classification and root-mean-squared error (RMSE) for regression) on each test set. Fig 3 contains the summary results for the 33 datasets. Fig 3A gives an overview of the predictive performance. To make the comparison easier, the median test scores are normalized to between 0 and 1 based on the minimum and maximum scores of each dataset. More details of the predictive performance results including boxplots of scores for all tasks and precision-recall curves for all classification tasks are provided in Figs B and C in S3 Appendix. Moreover, we perform a Wilcoxon signed-rank test [57] on the test scores and the percentage of times a method is significantly outperformed by another is given in Fig 3B. Lastly, run times of each method on the 33 datasets are shown in Fig 3C.

On all datasets `KernelBiome` achieves the best or close to best performance and (almost) always captures useful information (green labels in Fig 3A), indicating that the proposed procedure is well-adapted to microbiome data. The kernel which was selected most often by `KernelBiome` and the frequency with which it was selected are shown in Table B in S3 Appendix. There are several interesting observations: (1) Even though `KernelBiome` selects mostly the Aitchison kernel on rmp, it outperforms `KB-Aitchison`, we attribute this to the advantage of the data-driven zero-imputation. (2) On datasets were the top kernel is selected

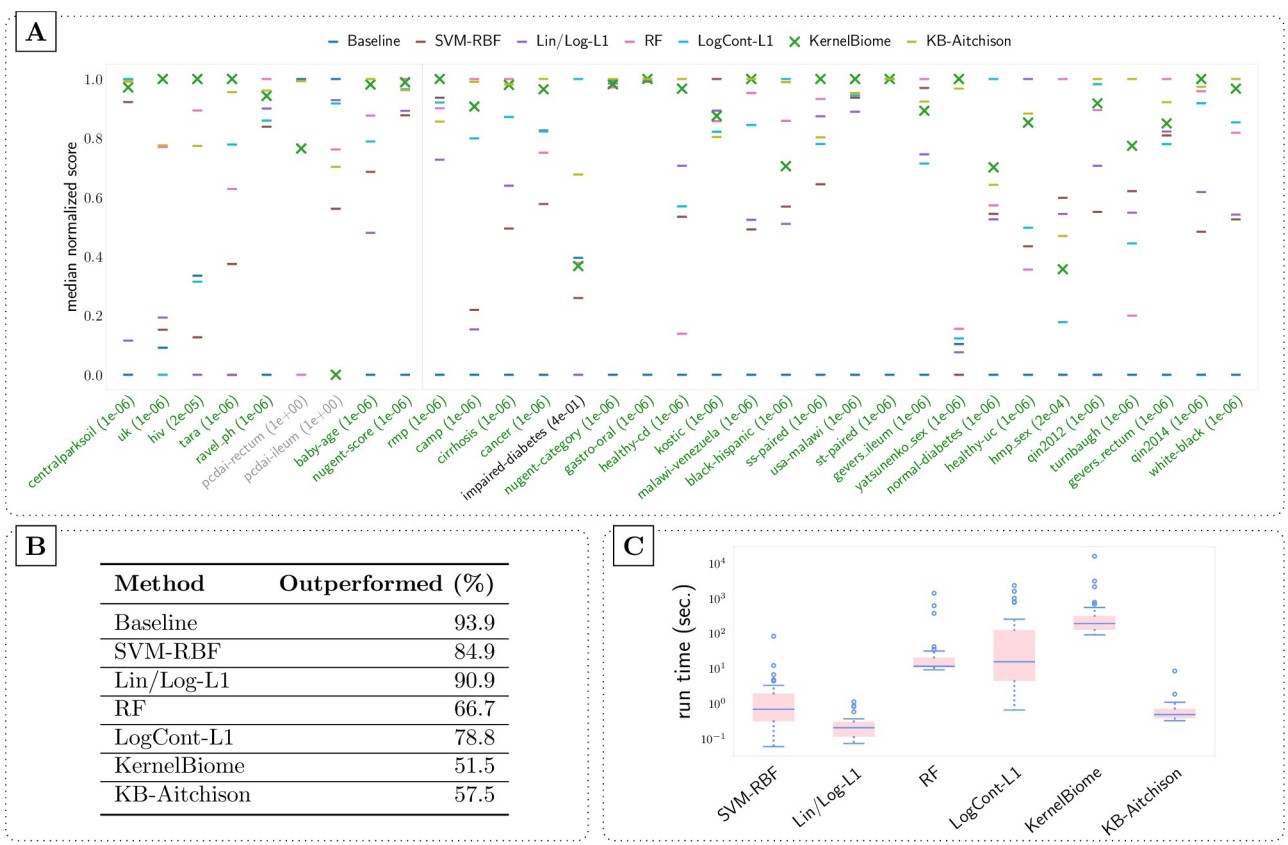

**Fig 3.** (A) Comparison of predictive performance on 33 public datasets (9 regression and 24 classification tasks, separated by the grey vertical line in the figure) based on 20 random 10-fold CV. On the two datasets with grey tick labels no method significantly outperforms the baseline based on the Wilcoxon signed-rank test, meaning that there is little signal in the data. The ones in green are the datasets where `KernelBiome` significantly outperforms the baseline, while it does not on the single dataset with the black label. The corresponding p-values are provided in brackets. (B) Percentage of time a method is significantly outperformed by another based on the Wilcoxon signed-rank test. (C) Average run time of each method on each dataset. A significance level of 0.05 is used.

consistently (e.g., uk, hiv and tara) `KernelBiome` generally performs very well and in these cases strongly outperformed both log-contrast based methods `KB-Aitchison` and `LogCont-L1`. (3) The predictive performance is substantially different between `KB-Aitchison` and `LogCont-L1` which we see as an indication that the type of regularization (kernel-based vs $\ell^1$-regularization, respectively) is crucial in microbiome datasets.

## 3.2 Predictive performance given prior information

In many applications, in particular in biology, prior information about a system is available and should be incorporated into the data analysis. As we show in Sec. 2.2.2, `KernelBiome` allows for incorporating prior knowledge on the relation between individual components (e.g., taxa). We will illustrate in this section that given informative prior knowledge, the predictive performance of `KernelBiome` can be improved, while if the prior knowledge is uninformative or incorrect, the predictive performance can be harmed. We conduct a semi-synthetic experiment based on the uk dataset. The dataset has 327 species and $n = 882$ samples. We generate the response $Y$ based on two processes: (DGP1) a linear log-contrast model where species under phylum Bacteroidetes all have coefficient $\beta_B$, species under phylum Proteobacteria all have coefficient $\beta_P$, and all coefficients corresponding to other species are set to 0; (DGP2) a

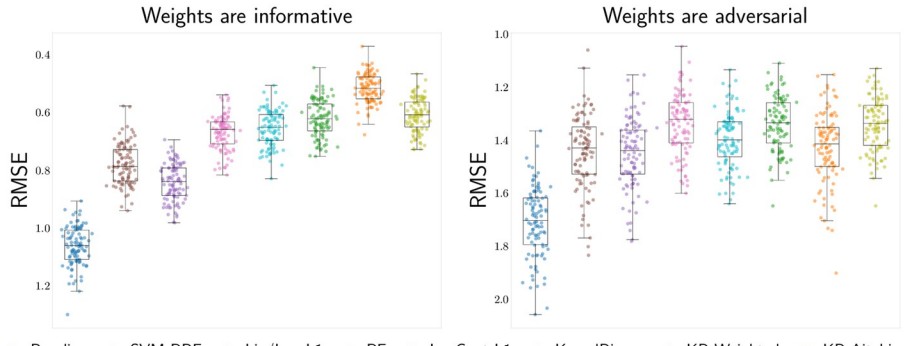

**Fig 4.** Left and middle: Predictive performance of weighted `KernelBiome` when the given weights are informative (DGP1) and adversarial (DGP2) based on 200 repetitions. Right: the number of other methods each method is significantly outperformed by based on Wilcoxon signed-rank test (siginificance level 0.05), under DGP1 and DGP2.

linear log-contrast model where the first half of the species under Bacteroidetes are given coefficient $-\beta_B$ while the second half are given coefficient $\beta_B$, similarly for Proteobacteria.

We construct a weight matrix based on the phylum each species belongs to (similar as in Prop. 2.2). By construction these weights are informative if the data are generated from DGP1, but not if the data are generated from DGP2. For each DGP, we sample 100 data points for training and another 100 data points for testing, repeated for 200 times. We compare the predictive performance of all methods mentioned in Sec. 3.1 and weighted `KernelBiome` (KB-Weighted) in Fig 4. The results show that when the weights are indeed informative, KB-Weighted achieves the best performance and is significantly better than the unweighted version (p-value $2.34 \times 10^{-18}$ based on Wilcoxon signed-rank test). On the other hand, when the weights do not align with the underlying generating mechanism, the predictive performance of KB-Weighted can be significantly worse than the unweighted one (p-value $1.25 \times 10^{-9}$ based on Wilcoxon signed-rank test). A table containing the number of other methods a method is significantly outperformed by under the two DGPs is also provided in Fig 4. All methods significantly outperform the baseline in this example, and the corresponding p-values are all below $2 \times 10^{-17}$.

### 3.3 Model analysis with KernelBiome

As shown in the previous section, `KernelBiome` results in fitted models with state-of-the-art prediction performance. This is useful because supervised learning procedures can be used in two types of applications: (1) To learn a prediction model that has a direct application, e.g., as a diagnostic tool, or (2) to learn a predictive model as an intermediate step of an exploratory analysis to find out what factors could be driving the response. As discussed above (2) requires us to take the compositional nature of the predictors into account to avoid misleading conclusions. We show how the `KernelBiome` framework can be used to achieve this based on two datasets: (i) *cirrhosis*, based on a study analyzing the differences in microbial compositions between $n = 130$ healthy and cirrhotic patients [58] and (ii) *centralparksoil*, based on a study analyzing the pH concentration using microbial compositions from $n = 580$ soil samples [59]. Our aim is not do draw novel biological conclusions, but rather to showcase how `KernelBiome` can be used in this type of analysis.

To reduce the complexity, we screen the data using `KernelBiome` with the Aitchison kernel and only keep the 50 taxa with the highest absolute CFIs. (As the analysis described here is only an illustration and we are not trying to compare methods, overfitting in this screening

step is not a concern. In practice, however, it may be relevant to validate the sensitivity of the results using for example subsampling). We then fit `KernelBiome` with default parameter grid. For cirrhosis this results in the Aitchison kernel and for centralparksoil in the Aitchison-RBF kernel. As outlined in Sec. 2.3.2, we can then apply a kernel PCA with a compositionally adjusted component influence. The result for centralparksoil is given in Fig 5A (for cirrhosis see Fig D in S3 Appendix). This provides some direct information on which perturbations affects each principle component (e.g., "[g]DA101[s]98" affects the first component the most positively and "Sphingobacterium[s]multivorum" affects the second component the most negatively). Moreover, it also directly provides a tool to detect groupings or outliers of the samples. For example, the samples in the top middle (i.e., center of the U-shape) in Fig 5A could be investigated further as they behave different to the rest.

A further useful quantity is the CFI, which for cirrhosis is given in Fig 5B (left) (for centralparksoil see Fig E in S3 Appendix). They explicitly take the compositional structure into account and have an easy interpretation. For example, "Prevotella_timonensis" has a CFI of 0.07 which implies that on average solely increasing "Prevotella_timonensis" will lead to a larger predicted response. We therefore believe that CFIs are more trustworthy than relying on for example Gini-importance for random forests, which does not have a clear interpretation due to the compositional constraint.

Lastly, one can also use the connection between kernels and distances to construct useful scalar summary statistics. In Fig 5B (right), we use the kernel-distance to the geometric median in the healthy subpopulation as a scalar indicator for the healthiness of the microbiome. In comparison with more standard scalar summary statistics such as the Simpson diversity, it is targeted to distinguish the two groups.

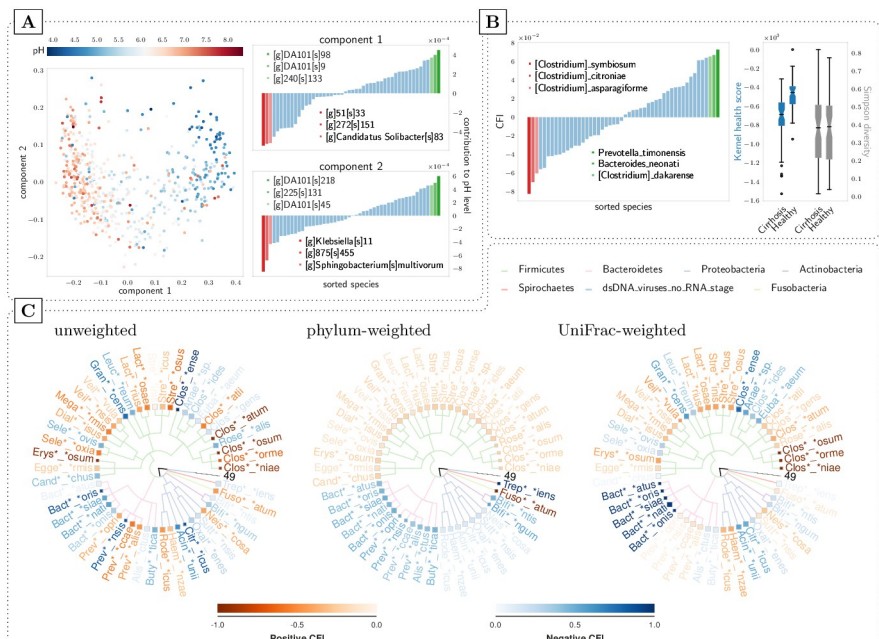

**Fig 5.** (A) shows a kernel PCA for the centralparksoil dataset with 2 principle components. On the right, the contribution of the species to each of the two components is given (see Sec. B of S5 Appendix for details). (B) and (C) are both based on the cirrhosis dataset. In (B) the CFI values are shown on the left and the right plot compares the proposed kernel health score with Simpson diversity. In (C) the scaled CFI values for are illustrated for different weightings. A darker color shade of the (shortened) name of the microbiota signifies a stronger (positive resp. negative) CFI.

### 3.4 Model analysis given prior information

Including prior information in `KernelBiome` can be used to improve the interpretation of the model analysis step. To illustrate how this works in practice, we again consider the screened cirrhosis dataset. We apply `KernelBiome` with an Aitchison-kernel and $c$ equal to half the minimum non-zero relative abundance without weighting, with a phylum-weighting and with a UniFrac-weighting. The resulting scaled CFI values for each are visualized in Fig 5C. The phylum-weighting corresponds to giving all taxa within the same phylum the same weights and the UniFrac-weighting is a weighting that incorporates the phylogenetic structure based on the UniFrac-distance and is described in Sec. C of S2 Appendix. As can bee seen in Fig 5C, the phylum weighting assigns approximately the same CFI to each variable in the same phylum, this is expected given that the phylum weighting has exactly the structure given in Proposition 2.2. Moreover, the UniFrac-weighting leads to CFI values that lie in-between the unweighted and phylum-weighted versions. Similar effects are seen for different kernels as well. The same plots for the generalized-JS kernel are provided in Fig G in S3 Appendix.

## 4 Discussion and conclusions

In this work, we propose the `KernelBiome` framework for supervised learning with compositional covariates consisting of two main ingredients: data-driven model selection and model interpretation. Our approach is based on a flexible family of kernels targeting the structure of microbiome data, and is able to work with different kernel-based algorithms such as SVM and kernel ridge regression. One can also incorporate prior knowledge, which is crucial in microbiome data analysis. We compare `KernelBiome` with other state-of-the-art approaches on 33 microbiome datasets and show that `KernelBiome` achieves improved or comparable results. Moreover, `KernelBiome` provides multiple ways to extract interpretable information from the fitted model. Two novel measures, CFI and CPD, can be used to analyze how each component affects the response. We prove the consistency of these two measures and illustrate them on simulated and real datasets. `KernelBiome` also leverages the connection between kernels and distances to conduct distance-based analysis in a lower-dimensional space.

## Supporting information

**S1 Appendix. Details on CFI and CPD.** Formal definitions of perturbations and estimators related to CFI and CPD.
(PDF)

**S2 Appendix. Details on kernels included in `KernelBiome`.** Overview of different kernel types, details on how they connect to distances and description of weighted kernels.
(PDF)

**S3 Appendix. Details and additional results for experiments in Sec. 3.** Datasets pre-processing, parameter setup, construction of the weighting matrices with UniFrac-distance and further experiment results based on the cirrhosis and centralparksoil datasets.
(PDF)

**S4 Appendix. Additional experiments with simulated data.** Consistency of CFI and CPD and comparison of CFI and CPD with their non-simplex counterparts.
(PDF)

**S5 Appendix. Background on kernels.** Mathematical background on kernels and details on dimensionality and visualization with kernels.
(PDF)

**S6 Appendix. Proofs.** Proof of theorems and propositions.
(PDF)

**S7 Appendix. List of kernels implemented in `KernelBiome`.**
(PDF)

## Acknowledgments

The authors would like to thank Christian Müller for detailed feedback and suggestions on this work, Johannes Ostner for help creating the circle plots, Jeroen Raes and Doris Vandeputte for making their raw data available.

## Author Contributions

**Conceptualization:** Shimeng Huang, Elisabeth Ailer, Niki Kilbertus, Niklas Pfister.

**Data curation:** Shimeng Huang, Elisabeth Ailer.

**Formal analysis:** Shimeng Huang, Elisabeth Ailer, Niki Kilbertus, Niklas Pfister.

**Funding acquisition:** Niki Kilbertus, Niklas Pfister.

**Investigation:** Shimeng Huang, Elisabeth Ailer.

**Methodology:** Shimeng Huang, Elisabeth Ailer, Niki Kilbertus, Niklas Pfister.

**Project administration:** Niki Kilbertus, Niklas Pfister.

**Resources:** Niki Kilbertus, Niklas Pfister.

**Software:** Shimeng Huang, Elisabeth Ailer, Niklas Pfister.

**Supervision:** Niki Kilbertus, Niklas Pfister.

**Validation:** Shimeng Huang, Elisabeth Ailer, Niki Kilbertus, Niklas Pfister.

**Visualization:** Shimeng Huang, Elisabeth Ailer.

**Writing – original draft:** Shimeng Huang, Elisabeth Ailer, Niki Kilbertus, Niklas Pfister.

**Writing – review & editing:** Shimeng Huang, Elisabeth Ailer, Niki Kilbertus, Niklas Pfister.

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
