## [Decision Letter · Decision Letter 0]

29 Mar 2023

Dear Ms. Huang,

Thank you very much for submitting your manuscript "Supervised learning and model analysis with compositional data" for consideration at PLOS Computational Biology.

As with all papers reviewed by the journal, your manuscript was reviewed by members of the editorial board and by several independent reviewers. In light of the reviews (below this email), we would like to invite the resubmission of a significantly-revised version that takes into account the reviewers' comments.

The reviewers both agree on the value of the study, an opinion we share, but request additional clarifications. We note that a common theme is both reviewers wish to see a clearer statement of how the proposed methods compared to others, which we agree is important. As an additional editorial request, we ask that the authors rework the figures to avoid the use of very small fonts (particularly Fig 1 [bottom right] and Fig 4).

We cannot make any decision about publication until we have seen the revised manuscript and your response to the reviewers' comments. Your revised manuscript is also likely to be sent to reviewers for further evaluation.

Sincerely,

Luis Pedro Coelho

Academic Editor

PLOS Computational Biology

Kiran Patil

Section Editor

PLOS Computational Biology

The reviewers both agree on the value of the study, an opinion we share, but request additional clarifications. We note that a common theme is both reviewers wish to see a clearer statement of how the proposed methods compared to others, which we agree is important. As an additional editorial request, we ask that the authors rework the figures to avoid the use of very small fonts (particularly Fig 1 [bottom right] and Fig 4).

Reviewer's Responses to Questions

**Comments to the Authors:**

Reviewer #1: The manuscript by Shimeng Huang et al. proposes KernelBiome, a kernel-based nonparametric regression and classification framework for compositional data. The method is specifically developed to deal with sparse compositional data and can incorporate prior knowledge in terms of phylogenetic structure. The algorithm is validated experimentally on 33 publicly available microbiome datasets and compared with state-of-the-art solutions. The code is available as an open-source python package.

The topic involved in the paper is suitable for publication in PLOS Computational Biology.

I find the methodological solution quite interesting. It is described in detail in both the main paper and the Supplementary Material. I have more comments about the experimental validation of the proposed solution:

1. As a general comment, the manuscript has a quite extensive supplementary material in terms of Appendix, while the main text is more limited, especially in terms of main Figures. I feel that some of the more important results/figures may be moved from the supplementary to the main text.

2. Following the previous point, the comparison among different classifiers is summarized in Figure 3 for only a fraction of the considered datasets (8 among 33 if I understand correctly). I think it would be important to have a figure here that summarizes the results for all considered datasets.

3. Despite the extensive validation, I still don't get to which extent the proposed solution outperforms the existing ones. For example, in how many cases/datasets the proposed method outperforms the other ones? Can you suggest (or not) your solution based on some data characteristics?

4. Could you add a statistical test to evaluate if differences in terms of accuracies are statistically significant?

5. I find interesting that the method can deal with prior information. Which is the added value of this information in terms of classification accuracies? I think it would be relevant performing a comparison in this direction (i.e., comparing results by incorporating or not the prior information).

6. It is not very clear to me which are the free parameters that should be set for the proposed solution. In case, a sensitivity analysis to them should be performed.

7. What in terms of computational complexity of the proposed solution with respect to the compared ones? Please add some empirical evaluations.

Reviewer #2: In this manuscript, Huang et al., propose a kernel-based nonparametric regression and classification framework to address the challenges of compositionality and sparsity in analyzing compositional data. The authors compared the proposed framework with existing methods on publicly available microbiome datasets. Overall, I found the paper to be well-written and well-organized. I have some comments for the authors.

1. Accuracy and MSE are applied in the classification and regression tasks, respectively. In classification tasks with unbalanced data, I don't think accuracy and AUROC are appropriate metrics. Therefore, I am highly concerned with the performance presented in the manuscript. Moreover, MSE and RMSE are both commonly used metrics for evaluating the performance of regression models. MSE is in square units of the target variable, which can make it difficult to interpret the results.

2. Line 357; Please consider add more details on the baseline.

3. Line 373-375: The authors claimed that "On all datasets KernelBiome achieves the best or close to best performance, indicating that the proposed procedure is well-adapted to microbiome data." However, this is not true at all when you look at Figure 6. In some case, the performance of KernelBiome is almost the worst.

**Have the authors made all data and (if applicable) computational code underlying the findings in their manuscript fully available?**

Reviewer #1: Yes

Reviewer #2: Yes

PLOS authors have the option to publish the peer review history of their article (what does this mean?). If published, this will include your full peer review and any attached files.

Reviewer #1: No

Reviewer #2: No
---

## [Decision Letter · Decision Letter 1]

24 May 2023

Dear Dr. Huang,

Thank you very much for submitting your manuscript "Supervised learning and model analysis with compositional data" for consideration at PLOS Computational Biology. As with all papers reviewed by the journal, your manuscript was reviewed by members of the editorial board and by several independent reviewers. The reviewers appreciated the attention to an important topic. Based on the reviews, we are likely to accept this manuscript for publication, providing that you modify the manuscript according to the review recommendations.

All scientific questions have been address. We agree with reviewer #2 that adding the p-value to the figures is best practices.

Sincerely,

Luis Pedro Coelho

Academic Editor

PLOS Computational Biology

Kiran Patil

Section Editor

PLOS Computational Biology

All scientific questions have been address. We agree with reviewer #2 that adding the p-value to the figures is best practices.

Reviewer's Responses to Questions

**Comments to the Authors:**

Reviewer #1: I feel the authors have answered positively to my previous comments and modified the manuscript according to them.

Reviewer #2: All my comments have been adequately addressed. On another note, the statistical P value should present with all the figures when comparing different methods. By including P-values, readers can better assess the significance of the findings and make informed interpretations.

**Have the authors made all data and (if applicable) computational code underlying the findings in their manuscript fully available?**

Reviewer #1: Yes

Reviewer #2: Yes

PLOS authors have the option to publish the peer review history of their article (what does this mean?). If published, this will include your full peer review and any attached files.

Reviewer #1: No

Reviewer #2: No

Figure Files:

Data Requirements:

Reproducibility:

References:

---

## [Editor Report · Decision Letter 2]

3 Jun 2023

Dear Dr. Huang,

We are pleased to inform you that your manuscript 'Supervised learning and model analysis with compositional data' has been provisionally accepted for publication in PLOS Computational Biology.

Best regards,

Luis Pedro Coelho

Academic Editor

PLOS Computational Biology

Kiran Patil

Section Editor

PLOS Computational Biology

---

## [Editor Report · Acceptance letter]

27 Jun 2023

PCOMPBIOL-D-23-00094R2 

Supervised learning and model analysis with compositional data

Dear Dr Huang,

I am pleased to inform you that your manuscript has been formally accepted for publication in PLOS Computational Biology. Your manuscript is now with our production department and you will be notified of the publication date in due course.

With kind regards,

Zsuzsanna Gémesi
